# Assessing the applicability of the new Global Lung Function Initiative reference values for the diffusing capacity of the lung for carbon monoxide in a large population set

**Pierre-Marie Wardyn**[1⊚], **Virginie de Broucker**[1,2⊚], **Cécile Chenivesse**[3,4,5‡], **Annie Sobaszek**[2,6‡], **Richard Van Bulck**[1‡], **Thierry Perez**[1,4⊚], **Jean-Louis Edmé**[1,2⊚], **Sébastien Hulo**[1,2⊚]*

1 Service des Explorations Fonctionnelles Respiratoires, CHU Lille, Lille, France, 2 EA 4483—IMPECS—IMPact de l'Environnement Chimique sur la Santé humaine, Univ. Lille, Lille, France, 3 Service de Pneumologie et Immuno-Allergologie, Centre de Référence Constitutif des Maladies Pulmonaires Rares, CHU Lille, Lille, France, 4 INSERM U1019—CNRS UMR 8204, Institut Pasteur de Lille—CIIL—Center for Infection and Immunity of Lille, Lille, France, 5 Univ. Lille, Lille, France, 6 Service de Médecine du Travail, CHU Lille, Lille, France

⊚ These authors contributed equally to this work.
‡ These authors also contributed equally to this work.
* sebastien.hulo@univ-lille.fr

## Abstract

### Background

The single-breath diffusing capacity of the lung for carbon monoxide ($D_{LCO}$) interpretation needs the comparison of measured values to reference values. In 2017, the Global Lung Function Initiative published new reference values (GLI-2017) for $D_{LCO}$, alveolar volume ($V_A$) and transfer coefficient of the lung for carbon monoxide ($K_{CO}$). We aimed to assess the applicability of GLI-2017 reference values for $D_{LCO}$ on a large population by comparing them to the European Community of Steel and Coal equations of 1993 (ECSC-93) widely used.

### Methods

In this retrospective study, spirometric indices, total lung capacity, $D_{LCO}$, $V_A$ and $K_{CO}$ were measured in adults classified in 5 groups (controls, asthma, chronic bronchitis, cystic fibrosis, and interstitial lung diseases (ILD)). Statistical analysis comparing the 2 equations sets were stratified by sex.

### Results

4180 tests were included. GLI-2017 z-scores of the 3 $D_{LCO}$ indices of the controls (n = 150) are nearer to 0 (expected value in a normal population) than ECSC-93 z-scores. All groups combined, in both genders, $D_{LCO}$ GLI-2017 z-scores and %predicted are significantly higher than ECSC z-scores and %predicted. In the ILD group, differences between the 2 equation sets depend on the $D_{LCO}$ impairment severity: GLI-2017 z-scores are higher than ECSC z-

**Data Availability Statement:** Please note that we cannot share the de-identified data set for legal reasons on the Research Ethics Board of the

University Hospital of Lille demand. The data set includes sensitive medical information about included patients as medical history and PFT results. Sex, age, height and weight are also included in the data set, which can potentially allow the patients' identification. Data requests may be sent to the Research Ethics Board of the University Hospital of Lille (contact: +33 3.20.44.41.65 / cppnordouestiv@univ-lille2.fr).

**Funding:** The authors received no specific funding for this work.

**Competing interests:** The authors have declared that no competing interests exist.

scores in patients with no or "mild" decrease in $D_{LCO}$, but are lower in "moderate" or "severe" decrease.

## Conclusion

GLI-2017 reference values for $D_{LCO}$ are more suitable to our population and influence the diagnostic criteria and severity definition of several lung diseases.

## Introduction

The single-breath diffusing capacity of the lung for carbon monoxide ($D_{LCO}$) is a simple non-invasive way to evaluate the alveolar-capillary gas exchanges [1]. $D_{LCO}$ is a key element in the diagnosis and follow-up of diseases in which lung gas transfer is altered by an alveolar-capillary membrane damage, as seen in interstitial lung diseases.

$D_{LCO}$ measured values need to be compared to reference values calculated with equations based on age, sex and height. Nowadays, in Europe, the European Community of Steel and Coal (ECSC) reference values for $D_{LCO}$ parameters, published in 1983 [2] (ECSC-83) and updated in 1993 [3,4] (ECSC-93), are the most commonly used. They are linear regression equations based on regression equations collation published before 1983. They are suitable for European women and men from 25 to 70 years old. Between 18 and 25 years old, predicted values are based on an age fixed at 25 years old. Before 18 years old, several equations sets can be applied including those of Polgar and Promadhat published in 1971 [5]. Besides the discontinuity problem between adults and children, the methods employed to produce those equations are far from current standards.

In 2017, the Global Lung Function Initiative (GLI) published new reference values for $D_{LCO}$, alveolar volume ($V_A$) and transfer coefficient of the lung for carbon monoxide ($K_{CO}$) for Caucasians aged from 5 to 85 years old (GLI-2017) [6]. Data derived from measured values in 9170 subjects from 14 countries excluding France due to French ethical laws.

Before using a set of equations on daily basis, the most appropriate should be chosen after comparison to the different available options [7,8]. To date, two articles have been published with this aim for GLI-2017 $D_{LCO}$ equations. The first on 145 patients with idiopathic pulmonary fibrosis evaluating the impact of GLI-2017 on clinical trial eligibility for those patients [9], and the second evaluating the effect of GLI-2017 on the normal/abnormal classification of $D_{LCO}$ results [10]. Even if they both concluded that the GLI-2017 should be adopted in laboratories, the need for a study including controls and multiple categories of patients with several degrees of decrease in $D_{LCO}$ still remains.

In our study, we aimed to assess the applicability of the GLI-2017 reference values for $D_{LCO}$ in adults by comparing them to the ECSC-93 equations currently used.

## Materials and methods

### Study population

We selected data of a control group and patients with symptoms compatible with one of four diseases (asthma, chronic bronchitis, cystic fibrosis, and interstitial lung diseases (ILD)) from pulmonary function tests (PFT) performed on daily basis between November 15[th] 2012 and May 30[th] 2016 in the PFT laboratory of the university hospital of Lille, France. PFTs providing all the following indices were selected: forced vital capacity (FVC), forced expiratory volume

in 1 second ($FEV_1$), $FEV_1$ to FVC ratio ($FEV_1/FVC$), total lung capacity (TLC), $D_{LCO}$, $V_A$ and $K_{CO}$. Only measurements performed before bronchodilation were included.

Exclusion criteria were: patients under 18 years old; patients classified in more than one group; missing data or outliers for sex, weight or height; any PFT other than spirometry with $D_{LCO}$ measurement; $D_{LCO}<1$mmol.$min^{-1}$.$kPa^{-1}$ (minimal value for the $D_{LCO}$ predicted value calculation [6,11]).

For each test, patient's characteristics (sex, age, height, weight), PFTs indices values and patient's medical history were recorded in the laboratory database. For the tests' inclusion, data from the PFT performed between November 15th 2012 and May 30th 2016, and including the indices mentioned above, were extracted to a Microsoft Excel® spreadsheet by the laboratory engineer. Then tests meeting exclusion criteria were excluded using the statistical software SAS® (version 9.4; Statistical Analysis System).

This study was performed in accordance with the Declaration of Helsinki. This human study was approved by the Research Ethics Board of the University Hospital of Lille (Comité de Protection des Personnes Nord Ouest IV)—approval: HP20/02. All adult participants provided written informed consent for the use of their data in the research field. Included patients were addressed to our department for routine functional evaluation.

## PFT measurements and results reports

Spirometry and lung volumes measured by body plethysmography and/or helium dilution were performed on a JAEGER® MasterScreen Body device (CareFusion, Hoechberg, Germany). Helium dilution was used for lung volumes measurements when body plethysmography was not feasible. $D_{LCO}$, $V_A$ and $K_{CO}$ measurements were performed on a JAEGER® MasterScreen PFT device (CareFusion, Hoechberg, Germany). Only test results meeting criteria for acceptability and reproducibility of the 2005 American Thoracic Society/European Respiratory Society recommendations (ATS/ERS-2005) [12–14] were included. In addition to the daily calibrations and verifications recommended by the manufacturer, the laboratory engineer (healthy and non-smoker) carried out a CO uptake measurement (biological calibration) every week. A $D_{LCO}$ variability above 10% led to a device verification.

ECSC-93 were applied to all concerned indices (FVC, $FEV_1$, $FEV_1/FVC$ and TLC) [2], the 2012 GLI equations for spirometry (GLI-2012) to FVC, $FEV_1$ and $FEV_1/FVC$ ratio [15] and ECSC-93 to $D_{LCO}$ [2]. The 1993 ECSC update recommend to calculate the $K_{CO}$ predicted value as the predicted $D_{LCO}$ to predicted TLC ratio [4], so the predicted $V_A$ used in this study is the predicted TLC for the $K_{CO}$ calculation according to ECSC. GLI-2017 were applied to $D_{LCO}$, $V_A$ and $K_{CO}$ [11].

Spirometric and TLC values were expressed as z-scores, $D_{LCO}$ and $V_A$ results as z-scores and %predicted, $K_{CO}$ adjustment on the ECSC-1993 predicted value as %predicted only (the standard deviations required for z-scores is not available).

## Groups creation

Selected PFTs came from 4 groups of patients with symptoms compatible with one of the following diseases: asthma, chronic bronchitis, cystic fibrosis, and ILD. The chronic bronchitis group included patients with symptoms compatible with chronic bronchitis (with a $FEV_1/FVC$ ratio $\geq 0.7$) or with chronic obstructive pulmonary disease (COPD) (with a $FEV_1/FVC$ ratio $< 0.7$ according to the GOLD 2021 report [16]). ILD category covers more than 150 entities such as sarcoidosis, idiopathic pulmonary fibrosis or pneumoconiosis [17]. The disease was initially filled out by the prescribing physician and recorded in the laboratory database at the time of the PFT validation. In order to ensure data quality, diagnoses were checked from

patient files in random samples of each groups (see S1 Appendix and S1 Table in S1 Appendix). Selected PFTs could have been performed at any stage of the concerned disease, from initial evaluation to advanced stage. The control group consisted of subjects complaining of dyspnea without parenchymal lung abnormalities and subjects from a health monitoring checking their work ability. Employees with professional exposures were excluded. Subject selection for this control group was performed independently from the PFT results.

## Statistical analysis

Data analysis was performed using the statistical software SAS® (version 9.4; Statistical Analysis System) and R (version 3.6.1; R Foundation, www.r-project.org). Five types of ventilatory disorders were created from spirometric and TLC values: "obstruction" ($FEV_1$/FVC below lower limit of normal (LLN) (z-score $<-1.645$), TLC $\geq$LLN), "restriction" (TLC $<$LLN, $FEV_1$/FVC $\geq$LLN), "mixed" (coexistence of obstruction and restriction) [18], Preserved Ratio Impaired Spirometry (PRISm) ($FEV_1$/FVC and TLC $\geq$LLN, $FEV_1$ $<$LLN) [19], "normal" (FVC, $FEV_1$, $FEV_1$/FVC and TLC $\geq$LLN) and "other" (not elsewhere classified).

$D_{LCO}$ parameters below LLN were considered altered. The ATS/ERS-2005 degrees of $D_{LCO}$ impairment were used: "Mild" ($D_{LCO}$>60%predicted and <LLN), "Moderate" ($D_{LCO}$>40% and $\leq$60%predicted), "Severe" ($D_{LCO}\leq$40%predicted) [18].

Data analysis were stratified by sex and reported by subject groups. Continuous quantitative variables were expressed as medians, 1st and 3rd quartiles; qualitative variables as absolute values and frequencies (percentages).

Non-parametric Wilcoxon and Kruskall-Wallis tests were used for data without normal distribution to compare quantitative variables between groups, Chi-squared tests to compare qualitative variables, Wilcoxon signed-rank tests to compare PFT values between equation sets. Differences in $D_{LCO}$ and $V_A$ alteration prevalence rates according to each equation sets were tested using McNemar's Chi-squared test.

Linear or multilinear regressions were used to study the relationship between the differences of the 2 $D_{LCO}$ z-scores, calculated with each equation sets, and age, sex, and decrease severity.

Statistical significance was set at $p < 0.05$ for all tests.

## Results

From the 15th November 2012 to the 30 May 2016, 9699 tests with spirometry and $D_{LCO}$ measurements were performed. 4180 tests were included (representing 2898 subjects) with 2181 tests in men and 1999 in women (Fig 1).

Table 1 shows the main characteristics of our population. Ventilatory defect types are significantly different between groups (Table 1). In the control group, "normal" ventilatory profiles proportions are close to the 95% expected in a normal population.

GLI-2012 produce significantly lower median z-scores for $FEV_1$ ($p < 0.0001$ in both genders), FVC ($p < 0.0001$ in both genders) and $FEV_1$/FVC ($p < 0.0001$ in both genders) than ECSC, all groups combined, in both genders (Table 2). In control subjects, ECSC $FEV_1$ and FVC median z-scores are more distant from the zero expected value than GLI-2012 values.

For both equation sets, $D_{LCO}$, $V_A$ and $K_{CO}$ z-score and %predicted are statistically linked to subject groups ($p < 0.0001$ in both genders; Kruskall-Wallis tests). Median z-scores in the control group are close to the zero expected with GLI-2017 in both genders (Table 3). In this group, standard deviations are also slightly closer to zero using GLI-2017 (0.819 in women, 0.928 in men) than using ECSC (0.866 in women, 0.968 in men). Furthermore, in this group, z-score differences between the 2 equations sets are linearly correlated to age ($p < 0.0001$) and

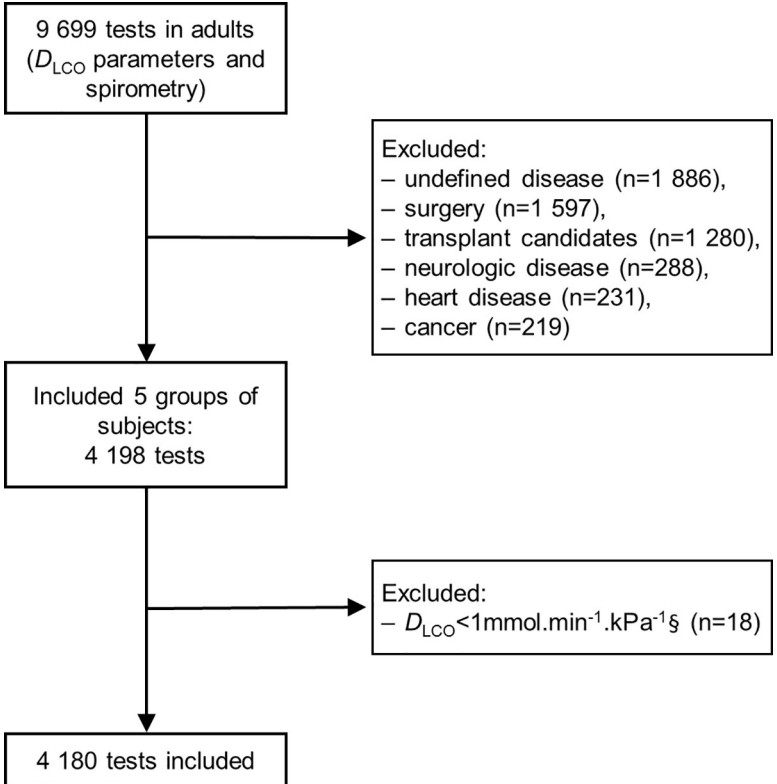

**Fig 1. Study flow-chart.** $D_{LCO}$: Diffusing capacity of the lung for carbon monoxide. §: Minimal value for the $D_{LCO}$ predicted value calculation [6,11].

height (p = 0.0004 in women, p = 0.0392 in men) in both genders, with a stronger trend in women. In the asthma and cystic fibrosis groups, GLI-2017 produce significantly higher $D_{LCO}$ z-scores than ECSC in both genders (Asthma group: p < 0.0001 in both genders. Cystic fibrosis group: p < 0.0001 in both genders). In the chronic bronchitis and ILD groups, the same trends are seen only in women, with a significant difference between $D_{LCO}$ z-scores in the ILD group only (p < 0.0001 in the ILD group; p = 0.27 in the chronic bronchitis group). In men with chronic bronchitis or ILD, which are the groups with the lowest $D_{LCO}$ values, GLI-2017 generate significantly lower $D_{LCO}$ z-scores than ECSC (p < 0.0001 in the chronic bronchitis group; p < 0.0001 in the ILD group).

For %predicted $D_{LCO}$, GLI-2017 results are significantly higher than ECSC values (p < 0.0001 in men and women). In the control group, $D_{LCO}$ impairment prevalence rates are nearer to the expected 5% using GLI-2017 (6.5% in men, 1.4% in women) compared to ECSC (19.5% in men, 31.5% in women) (Table 4). In the other groups, $D_{LCO}$ impairment prevalence rates are significantly lower using GLI-2017 than using ECSC with larger differences in women (in women: p < 0.0001 in the chronic bronchitis group, p < 0.0001 in the cystic fibrosis group, p < 0.0001 in the ILD group; in men: p < 0.0001 in the chronic bronchitis group, p = 0.0026 in the cystic fibrosis group, p < 0.0001 in the ILD group). In the asthma group, for example, the proportion of $D_{LCO}$ measurements categorized as normal with GLI-2017 is significantly higher than using ECSC, especially in women (p < 0.0001 in both genders).

GLI-2017 and ECSC $D_{LCO}$ z-scores comparison by degree of decrease in $D_{LCO}$ according to ATS/ERS-2005 in the ILD group is shown in Fig 2. GLI-2017 $D_{LCO}$ z-scores are -2.03, -3.39, -5.47 in men and -1.55, -3.36, -7.22 in women, in "mild", "moderate" and "severe" categories

**Table 1. Characteristics of the population by sex.**

| Variables | Overall | Controls | Asthma | Chronic bronchitis | Cystic fibrosis | ILD | Statistics between groups p-value |
|---|---|---|---|---|---|---|---|
| | n = 4180 | n = 150 | n = 527 | n = 732 | n = 145 | n = 2626 | |
| **Male** | | | | | | | |
| Subjects, n | 2181 | 77 | 197 | 534 | 70 | 1303 | |
| Age, years | 59 (47; 68) | 40 (27; 51) | 46 (34; 56) | 59 (51; 66) | 28 (24; 37) | 62 (52; 70) | < 0.0001 |
| Weight, kg | 80 (70; 90) | 81 (72; 91) | 78 (69; 90) | 75 (66; 90) | 64 (57; 77) | 81 (72; 91) | < 0.0001 |
| Height, cm | 173 (169; 178) | 178 (174; 184) | 176 (171; 180) | 172 (168; 178) | 171 (168; 176) | 173 (168; 178) | < 0.0001 |
| BMI, kg.m$^{-2}$ | 26.3 (23.2; 29.8) | 25.9 (23.1; 28.1) | 25.1 (22.5; 28.4) | 25.3 (22.0; 29.4) | 21.7 (19.9; 24.9) | 27.2 (24.1; 30.4) | < 0.0001 |
| Ventilatory defect type | 2167 | 73 | 189 | 532 | 69 | 1300 | < 0.0001 |
| Normal | 658 (30.4%) | 73 (94.8%) | 83 (43.9%) | 80 (15.0%) | 22 (31.9%) | 400 (30.8%) | |
| Restriction | 793 (36.6%) | 2 (2.6%) | 18 (9.5%) | 36 (6.8%) | 6 (8.7%) | 731 (56.2%) | |
| Obstruction | 361 (16.7%) | 2 (2.6%) | 55 (29.1%) | 234 (44.0%) | 12 (17.4%) | 58 (4.5%) | |
| Mixed | 322 (14.9%) | 0 | 26 (13.8%) | 168 (31.6%) | 29 (42%) | 99 (7.6%) | |
| PRISm | 31 (1.4%) | 0 | 7 (3.7%) | 14 (2.6%) | 0 | 10 (0.8%) | |
| Other | 2 (0.1%) | 0 | 0 | 0 | 0 | 2 (0.1%) | |
| Missing data‡ | 14 | 0 | 8 | 2 | 1 | 3 | |
| **Female** | | | | | | | |
| Subjects, n | 1999 | 73 | 330 | 198 | 75 | 1323 | |
| Age, years | 53 (41; 65) | 43 (29; 54) | 48 (34; 57) | 55 (48; 65) | 31 (22; 42) | 56 (43; 66) | < 0.0001 |
| Weight, kg | 67 (57; 80) | 63 (56; 78) | 67 (58; 79) | 62 (53; 76) | 53 (50; 60) | 68 (58; 82) | < 0.0001 |
| Height, cm | 162 (157; 167) | 163 (160; 167) | 162 (157; 168) | 161 (157; 166) | 164 (160; 167) | 161 (156; 166) | 0.00022 |
| BMI, kg.m$^{-2}$ | 25.5 (21.8; 30.8) | 23.7 (21.3; 28.9) | 25.3 (21.3; 29.7) | 23.4 (20.7; 28.4) | 20.2 (18.7; 21.8) | 26.4 (22.5; 31.5) | < 0.0001 |
| Ventilatory defect type | 1963 | 70 | 301 | 194 | 75 | 1323 | < 0.0001 |
| Normal | 1104 (56.2%) | 66 (94.3%) | 173 (57.5%) | 32 (16.5%) | 25 (33.3%) | 808 (61.1%) | |
| Restriction | 393 (20.2%) | 0 | 15 (5%) | 6 (3.1%) | 6 (8%) | 366 (27.7%) | |
| Obstruction | 294 (15.0%) | 2 (2.9%) | 76 (25.2%) | 110 (56.7%) | 20 (26.7%) | 86 (6.5%) | |
| Mixed | 118 (6.0%) | 0 | 27 (9%) | 40 (20.6%) | 21 (28%) | 30 (2.3%) | |
| PRISm | 51 (2.6%) | 1 (1.4%) | 9 (3.0%) | 6 (3.1%) | 3 (4%) | 32 (2.4%) | |
| Other | 3 (0.1%) | 2 (1.4%) | 1 (0.3%) | 0 | 0 | 1 (0.1%) | |
| Missing data‡ | 36 | 3 | 29 | 4 | 0 | 0 | |

Ventilatory defect types according to the 2005 recommendations for lung function test interpretation from the American Thoracic Society (ATS) and the European Respiratory Society (ERS): Obstruction: z-score of $FEV_1$/FVC ratio < -1.645 and TLC z-score ≥ -1.645; restriction: TLC z-score < -1.645 and z-score of $FEV_1$/FVC ratio ≥ -1.645; mixed (coexistence of obstruction and restriction); [18] PRISm (Preserved Ratio Impaired Spirometry): z-score of $FEV_1$/FVC ratio ≥ -1.645, TLC z-score ≥ -1.645 and $FEV_1$ z-score < -1.645; [19] normal: z-score of FVC, $FEV_1$, $FEV_1$/FVC ratio and TLC ≥ -1.645; other: Not elsewhere classified. ILD: Interstitial lung disease; BMI: Body mass index; $FEV_1$: Forced expiratory volume in 1 s; FVC: Forced vital capacity; TLC: Total lung capacity. Data are presented as n, median (Q1; Q3) and n (%). Kruskall-Wallis tests were used to compare qualitative variables between disease groups, Chi-squared tests to compare qualitative variables between disease groups. ‡Missing data only concern TLC.

respectively. GLI-2017 z-scores are significantly higher than ECSC z-scores in patients with no (p < 0.0001 in both genders) or "mild" decrease in $D_{LCO}$ (p < 0.0001 in both genders), but are significantly lower in "moderate" (p < 0.0001 in both genders) or "severe" (p < 0.0001 in both genders) decrease in $D_{LCO}$. The multivariate analysis confirmed that the z-scores differences (ECSC vs GLI-2017 z-scores) is influenced by the DLCO severity grade (taking age in account). Multiple linear regression analysis showed, in both genders, a highly significant

**Table 2. Z-scores of FVC, FEV₁, FEV₁/FVC and TLC according to ECSC-93 and GLI-2012 (except for TLC) by sex.**

| Variables | | Overall | Controls | Asthma | Chronic bronchitis | Cystic fibrosis | ILD |
|---|---|---|---|---|---|---|---|
| | | n = 4180 | n = 150 | n = 527 | n = 732 | n = 145 | n = 2626 |
| **Male** | | | | | | | |
| Tests, n | | 2181 | 77 | 197 | 534 | 70 | 1303 |
| FVC | ECSC-93 | -0.759 (-1.77; 0.19) | 0.88 (0.08; 1.35) | -0.13 (-1.29; 0.64) | -0.94 (-1.97; 0.09) | -0.67 (-2.18; 0.09) | -0.87 (-1.80; 0.05) |
| | GLI-2012 | -1.18 (-2.21; -0.26) | 0.23 (-0.46; 0.65) | -0.58 (-1.61; 0.08) | -1.33 (-2.34; -0.40) | -1.15 (-2.62; -0.36) | -1.25 (-2.24; -0.39) |
| FEV₁ | ECSC-93 | -1.29 (-2.47; -0.33) | 0.59 (-0.36; 1.17) | -1.31 (-2.47; -0.17) | -2.52 (-3.60; -1.41) | -2.19 (-4.22; -0.76) | -1.00 (-1.88; -0.21) |
| | GLI-2012 | -1.65 (-2.73; -0.68) | 0.16 (-0.77; 0.68) | -1.63 (-2.69; -0.54) | -2.79 (-3.73; -1.82) | -2.54 (-4.36; -0.89) | -1.37 (-2.24; -0.54) |
| FEV₁/FVC | ECSC-93 | -0.685 (-2.09; 0.40) | 0.002 (-0.48; 0.43) | -1.34 (-2.56; -0.32) | -2.75 (-4.42; -1.67) | -2.37 (-3.65; -0.74) | 0.05 (-0.84; 0.83) |
| | GLI-2012 | -0.910 (-2.19; 0.21) | -0.23 (-0.67; 0.38) | -1.58 (-2.56; -0.48) | -2.68 (-3.89; -1.83) | -2.55 (-3.51; -1.07) | -0.13 (-1.00; 0.58) |
| TLC | ECSC-93 | -1.70 (-3.06; -0.50) | -0.10 (-0.53; 0.63) | -0.77 (-1.56; 0.25) | -1.14 (-2.21; -0.14) | -1.65 (-2.60; -0.58) | -2.34 (-3.48; -1.16) |
| | Missing data | 14 (0.6%) | 0 | 8 (4.1%) | 2 (0.4%) | 1 (1.4%) | 3 (0.2%) |
| **Female** | | | | | | | |
| Tests, n | | 1999 | 73 | 330 | 198 | 75 | 1323 |
| FVC | ECSC-93 | -0.03 (-0.97; 0.83) | 0.91 (0.01; 1.69) | 0.13 (-0.75; 1.02) | -0.46 (-1.21; 0.41) | -0.99 (-2.33; 0.29) | -0.01 (-0.94; 0.80) |
| | GLI-2012 | -0.85 (-1.81; -0.09) | -0.04 (-0.64; 0.77) | -0.70 (-1.60; 0.21) | -1.28 (-2.07; -0.51) | -1.48 (-2.55; -0.48) | -0.86 (-1.81; -0.12) |
| FEV₁ | ECSC-93 | -0.62 (-1.73; 0.23) | 0.31 (-0.29; 1.08) | -0.65 (-1.81; 0.23) | -1.96 (-3.54; -1.11) | -1.92 (-4.10; -0.54) | -0.47 (-1.43; 0.30) |
| | GLI-2012 | -1.15 (-2.24; -0.28) | -0.042 (-0.74; 0.71) | -1.18 (-2.44; -0.26) | -2.70 (-3.76; -1.77) | -1.94 (-4.21; -0.88) | -1.03 (-1.98; -0.23) |
| FEV₁/FVC | ECSC-93 | -0.29 (-1.33; 0.55) | 0.04 (-0.60; 0.60) | -0.72 (-1.99; 0.02) | -2.52 (-4.90; -1.71) | -2.00 (-3.16; -0.52) | 0.03 (-0.68; 0.77) |
| | GLI-2012 | -0.52 (-1.43; 0.26) | -0.17 (-0.82; 0.33) | -0.97 (-1.98; -0.24) | -2.32 (-3.64; -1.75) | -1.98 (-2.81; -0.75) | -0.18 (-0.88; 0.50) |
| TLC | ECSC-93 | -0.72 (-1.68; 0.17) | 0.44 (-0.25; 0.99) | -0.23 (-1.09; 0.50) | -0.72 (-1.55; 0.21) | -1.30 (-2.40; -0.20) | -0.85 (-1.90; 0.01) |
| | Missing data | 36 (1.8%) | 3 (4.1%) | 29 (8.8%) | 4 (2.0%) | 0 | 0 |

ILD: Interstitial lung disease; FVC: Forced vital capacity; FEV₁: Forced expiratory volume in 1 s; TLC: Total lung capacity; ECSC-93: European Community for Steel and Coal 1993 reference values; [2,4] GLI-2012: Global Lung function Initiative 2012 reference values [15]. Data are presented as n, median z-score (Q1; Q3) and n (%).

relationship between z-scores differences (ECSC vs GLI-2017) and age (as in the "control" group) (p < 0.0001, in both genders), the decrease in $D_{LCO}$ severity (p < 0.0001, in both genders) and the interaction between age and severity (p < 0.0001 in both genders).

## Discussion

To our knowledge, this study is the first to apply $D_{LCO}$ measured values to the GLI-2017 reference values on a large population of patients and controls. Results in control group showed that the z-scores of the 3 $D_{LCO}$ indices are nearer to 0 (expected value in a normal population) according to GLI-2017 than with ECSC-93. Moreover, in this control population, the proportion of tests with altered indices was near the expected 5%. GLI-2017 equations increase the proportion of tests classified as normal for $D_{LCO}$ compared to ECSC, notably in the control and asthma groups. In the disease groups, we found a significant relationship between the severity degree of decrease in $D_{LCO}$ and the z-scores differences between GLI-2017 and ECSC. GLI-2017 produce lower $D_{LCO}$ z-scores for moderate to severe decrease in $D_{LCO}$ than ECSC, notably in the ILD group. In all groups, differences between the 2 equation sets were more pronounced in women. Finally, $K_{CO}$ results expressed as %predicted using GLI-2017 were significantly lower than using ECSC in men, but significantly higher in women.

We recruited subjects without a disease diagnosed by a physician. Ventilatory disorder types analysis showed a "normal" type proportion close to the 95% expected in a healthy population. To validate this control group, we applied the approach used to validate GLI-2012 [20,21]: z-scores close to zero, tests with a $D_{LCO}$ below LLN proportion. In this group, GLI-

**Table 3. Measured values, z-scores and %predicted for $D_{LCO}$, $V_A$ and $K_{CO}$ according to ECSC-93 and GLI-2017 by sex.**

| Variables | | Overall | Controls | Asthma | Chronic bronchitis | Cystic fibrosis | ILD |
|---|---|---|---|---|---|---|---|
| | | n = 4180 | n = 150 | n = 527 | n = 732 | n = 145 | n = 2626 |
| **Male** | | | | | | | |
| Subjects, n | | 2181 | 77 | 197 | 534 | 70 | 1303 |
| $D_{LCO}$, mmol.min$^{-1}$.kPa$^{-1}$ | | 5.64 (3.80; 8.13) | 10.08 (8.84; 11.42) | 9.01 (7.63; 10.05) | 5.20 (3.74; 6.70) | 9.01 (7.78; 10.46) | 5.08 (3.42; 7.06) |
| Z-score | ECSC-93 | -2.50 (-3.61; -1.41) | -0.95 (-1.48; 0.10) | -1.12 (-1.99; -0.16) | -2.67 (-3.80; -1.62) | -1.42 (-2.31; -0.55) | -2.76 (-3.83; -1.73) |
| | GLI-2017 | -2.45 (-4.05; -1.00) | -0.45 (-1.02; 0.43) | -0.64 (-1.50; 0.27) | -2.82 (-4.23; -1.48) | -0.67 (-1.68; 0.13) | -2.86 (-4.40; -1.52) |
| | ECSC vs GLI | p = 0.02 | p < 0.0001 | p < 0.0001 | p < 0.0001 | p < 0.0001 | p < 0.0001 |
| % pred. | ECSC-93 | 61.7 (43.9; 80.2) | 88.5 (80.6; 101.4) | 85.3 (73.4; 97.5) | 57.2 (41.7; 74.0) | 83.3 (71.5; 92.9) | 56.7 (40.1; 73.2) |
| | GLI-2017 | 64.7 (46.1; 85.1) | 93.9 (85.1; 107.2) | 90.9 (78.6; 104.6) | 60.0 (44.5; 78.4) | 91.0 (78.4; 101.8) | 59.5 (41.9; 77.4) |
| | ECSC vs GLI | p < 0.0001 | p < 0.0001 | p < 0.0001 | p < 0.0001 | p < 0.0001 | p < 0.0001 |
| $V_A$, L | | 5.16 (4.17; 6.12) | 6.65 (6.12; 7.51) | 6.24 (5.37; 6.70) | 5.48 (4.64; 6.15) | 5.06 (4.51; 5.94) | 4.75 (3.81; 5.73) |
| Z-score | ECSC-93 | -2.21 (-3.56; -1.05) | -0.40 (-1.03; .18) | -1.21 (-1.87; -0.24) | -1.68 (-2.77; -0.74) | -2.16 (-2.97; -1.01) | -2.86 (-3.96; -1.64) |
| | GLI-2017 | -1.26 (-2.60; -0.18) | 0.25 (-0.41; 0.63) | -0.24 (-1.05; 0.51) | -0.79 (-1.77; 0.12) | -1.26 (-2.25; -0.16) | -1.91 (-3.00; -0.71) |
| | ECSC vs GLI | p < 0.0001 | p < 0.0001 | p < 0.0001 | p < 0.0001 | p < 0.0001 | p < 0.0001 |
| % pred. | ECSC-93 | 76.8 (62.7; 89.2) | 93.9 (89.1; 99.5) | 87.8 (81.1; 97.6) | 82.2 (71.2; 92.0) | 76.9 (69.0; 87.8) | 69.9 (58.3; 83.5) |
| | GLI-2017 | 85.1 (69.6; 97.8) | 103.0 (95.4; 107.0) | 97.1 (88.4; 106.0) | 90.5 (79.5; 101.4) | 86.3 (76.0; 98.3) | 77.5 (65.2; 91.4) |
| | ECSC vs GLI | p < 0.0001 | p = 0.00039 | p < 0.0001 | p < 0.0001 | p < 0.0001 | p < 0.0001 |
| $K_{CO}$, mmol.min$^{-1}$.kPa$^{-1}$.L$^{-1}$ | | 1.12 (0.87; 1.40) | 1.45 (1.29; 1.61) | 1.43 (1.28; 1.65) | 0.97 (0.72; 1.25) | 1.70 (1.59; 1.91) | 1.08 (0.86; 1.34) |
| Z-score | GLI-2017 | -1.49 (-2.78; -0.44) | -0.50 (-1.03; -0.08) | -0.44 (-1.03; 0.38) | -2.23 (-3.63; -1.09) | 0.15 (-0.48; 1.00) | -1.60 (-2.77; -0.59) |
| % pred. | ECSC-93 | 82.4 (64.3; 97.4) | 94.3 (87.4; 100.3) | 97.1 (87.3; 107.5) | 71.5 (52.7; 88.2) | 102.4 (94.0; 115.0) | 80.9 (64.4; 96.1) |
| | GLI-2017 | 78.0 (60.7; 93.6) | 93.1 (85.2; 99.0) | 94.0 (85.2; 105.4) | 67.6 (49.7; 84.0) | 101.9 (93.8; 113.8) | 75.6 (60.7; 91.2) |
| | ECSC vs GLI | p < 0.0001 | p < 0.0001 | p < 0.0001 | p < 0.0001 | p < 0.0001 | p < 0.0001 |
| **Female** | | | | | | | |
| Subjects, n | | 1999 | 73 | 330 | 198 | 75 | 1323 |
| $D_{LCO}$, mmol.min$^{-1}$.kPa$^{-1}$ | | 5.44 (3.97; 6.67) | 6.99 (6.57; 7.81) | 6.45 (5.46; 7.41) | 4.54 (2.89; 5.91) | 6.40 (5.49; 7.53) | 5.06 (3.60; 6.31) |
| Z-score | ECSC-93 | -2.11 (-3.04; -1.27) | -1.27 (-1.74; -0.57) | -1.59 (-2.28; -0.87) | -2.56 (-3.64; -1.65) | -2.07 (-2.88; -1.08) | -2.27 (-3.17; -1.40) |
| | GLI-2017 | -1.46 (-3.11; -0.41) | -0.13 (-0.72; 0.42) | -0.67 (-1.48; 0.13) | -2.19 (-5.12; -1.03) | -0.96 (-1.68; 0.12) | -1.78 (-3.60; -0.66) |
| | ECSC vs GLI | p < 0.0001 | p < 0.0001 | p < 0.0001 | p = 0.27 | p < 0.0001 | p < 0.0001 |
| % pred. | ECSC-93 | 69.0 (54.1; 81.2) | 82.7 (76.9; 92.1) | 77.3 (68.1; 87.0) | 61.5 (40.6; 73.4) | 72.7 (62.8; 85.7) | 65.9 (50.4; 78.5) |
| | GLI-2017 | 79.3 (61.0; 93.6) | 97.8 (88.8; 107.1) | 89.6 (78.7; 102.1) | 69.6 (46.0; 84.4) | 86.2 (77.4; 101.8) | 74.8 (56.3; 89.8) |
| | ECSC vs GLI | p < 0.0001 | p < 0.0001 | p < 0.0001 | p < 0.0001 | p < 0.0001 | p < 0.0001 |
| $V_A$, L | | 4.12 (3.44; 4.73) | 4.85 (4.50; 5.37) | 4.51 (3.87; 5.02) | 4.06 (3.51; 4.69) | 4.13 (3.30; 4.65) | 4.01 (3.28; 4.58) |
| Z-score | ECSC-93 | -1.21 (-2.20; -0.30) | 0.17 (-0.54; 0.86) | -0.67 (-1.44; 0.06) | -1.36 (-2.36; -0.23) | -1.65 (-2.51; -0.78) | -1.40 (-2.38; -0.50) |
| | GLI-2017 | -1.04 (-2.18; -0.11) | 0.09 (-0.58; 0.63) | -0.49 (-1.33; 0.23) | -1.15 (-2.33; -0.07) | -1.61 (-2.76; -0.64) | -1.23 (-2.42; -0.29) |

(*Continued*)

**Table 3.** (Continued)

| Variables | | Overall | Controls | Asthma | Chronic bronchitis | Cystic fibrosis | ILD |
|---|---|---|---|---|---|---|---|
| | | n = 4180 | n = 150 | n = 527 | n = 732 | n = 145 | n = 2626 |
| | ECSC vs GLI | p < 0.0001 | p < 0.0001 | p < 0.0001 | p = 0.0031 | p = 0.66 | p = 0.00037 |
| % pred. | ECSC-93 | 85.1 (72.6; 96.1) | 98.7 (91.3; 106.1) | 91.8 (81.7; 100.8) | 82.5 (71.8; 96.1) | 80.7 (69.3; 90.7) | 82.8 (70.5; 93.7) |
| | GLI-2017 | 87.5 (75.2; 98.7) | 101.1 (93.1; 108.1) | 93.6 (84.8; 102.8) | 85.7 (73.2; 99.1) | 81.5 (71.2; 92.5) | 85.0 (72.8; 96.4) |
| | ECSC vs GLI | p < 0.0001 | p < 0.0001 | p < 0.0001 | p < 0.0001 | p < 0.0001 | p < 0.0001 |
| $K_{CO}$, mmol.min$^{-1}$.kPa$^{-1}$.L$^{-1}$ | | 1.32 (1.09; 1.52) | 1.43 (1.31; 1.60) | 1.45 (1.27; 1.63) | 1.11 (0.78; 1.42) | 1.62 (1.38; 1.88) | 1.28 (1.06; 1.47) |
| Z-score | GLI-2017 | -0.72 (-1.78; 0.18) | -0.16 (-0.90; 0.47) | -0.25 (-0.97; 0.60) | -1.57 (-3.77; -0.16) | 0.28 (-0.64; 1.22) | -0.88 (-1.96; 0.01) |
| % pred. | ECSC-93 | 80.9 (68.6; 92.6) | 85.0 (77.1; 94.7) | 85.6 (77.2; 96.4) | 70.0 (48.1; 87.6) | 90.7 (78.5; 104.0) | 79.4 (67.5; 90.9) |
| | GLI-2017 | 89.8 (75.5; 102.6) | 97.7 (87.3; 107.0) | 96.5 (86.6; 109.0) | 78.4 (52.9; 97.7) | 104.0 (91.0; 119.3) | 87.4 (73.5; 100.2) |
| | ECSC vs GLI | p < 0.0001 | p < 0.0001 | p < 0.0001 | p < 0.0001 | p < 0.0001 | p < 0.0001 |

ILD: Interstitial lung disease; $D_{LCO}$: Diffusing capacity of the lung for carbon monoxide; ECSC-93: European Community for Steel and Coal 1993 reference values; [2,4] GLI-2017: Global Lung function Initiative 2017 reference values; [6] % pred.: %predicted; $V_A$: Alveolar volume; $K_{CO}$: Transfer coefficient of the lung for carbon monoxide. Data are presented as n and median (Q1; Q3). Wilcoxon signed-rank tests were used. All $D_{LCO}$, $V_A$ and $K_{CO}$ values expressed as %pred. or z-scores were significantly different between groups of subjects using ECSC and GLI equations sets (p < 0.0001; Kruskall-Wallis tests).

**Table 4. Number of tests and prevalence rates for $D_{LCO}$, $V_A$ and $K_{CO}$ impairment according to ECSC-93 and GLI-2017, by sex.**

| Variables | | Overall | Controls | Asthma | Chronic bronchitis | Cystic Fibrosis | ILD |
|---|---|---|---|---|---|---|---|
| | | n = 4180 | n = 150 | n = 527 | n = 732 | n = 145 | n = 2626 |
| **Male** | | | | | | | |
| Subjects, n | | 2181 | 77 | 197 | 534 | 70 | 1303 |
| Abnormalities in $D_{LCO}$‡ | ECSC-93 | 1 507 (69.1%) | 15 (19.5%) | 67 (34%) | 397 (74.3%) | 29 (41.4%) | 999 (76.7%) |
| | GLI-2017 | 1 400 (64.2%) | 5 (6.5%) | 46 (23.4%) | 376 (70.4%) | 18 (25.7%) | 955 (73.3%) |
| | ECSC vs GLI | p < 0.0001 | p = 0.0044 | p < 0.0001 | p < 0.0001 | p = 0.0026 | p < 0.0001 |
| Abnormalities in $V_A$‡ | ECSC-93 | 1 360 (62.4%) | 9 (11.7%) | 62 (31.5%) | 271 (50.7%) | 43 (61.4%) | 975 (74.8%) |
| | GLI-2017 | 924 (42.4%) | 0 | 35 (17.8%) | 145 (27.1%) | 31 (44.3%) | 713 (54.7%) |
| | ECSC vs GLI | p < 0.0001 | p = 0.0077 | p < 0.0001 | p < 0.0001 | p = 0.0015 | p < 0.0001 |
| Abnormalities in $K_{CO}$‡ | GLI-2017 | 1013 (46.4%) | 7 (9.1%) | 29 (14.7%) | 334 (62.5%) | 2 (2.9%) | 641 (49.2%) |
| **Female** | | | | | | | |
| Subjects, n | | 1999 | 73 | 330 | 198 | 75 | 1323 |
| Abnormalities in $D_{LCO}$‡ | ECSC-93 | 1 282 (64.1%) | 23 (31.5%) | 152 (46.1%) | 150 (75.8%) | 43 (57.3%) | 914 (69.1%) |
| | GLI-2017 | 912 (45.6%) | 1 (1.4%) | 69 (20.9%) | 123 (62.1%) | 19 (25.3%) | 700 (52.9%) |
| | ECSC vs GLI | p < 0.0001 | p < 0.0001 | p < 0.0001 | p < 0.0001 | p < 0.0001 | p < 0.0001 |
| Abnormalities in $V_A$‡ | ECSC-93 | 758 (37.9%) | 6 (8.2%) | 69 (20.9%) | 84 (42.4%) | 38 (50.7%) | 561 (42.4%) |
| | GLI-2017 | 702 (35.1%) | 3 (4.1%) | 65 (19.7%) | 75 (37.8%) | 37 (49.3%) | 522 (39.5%) |
| | ECSC vs GLI | p < 0.0001 | p = 0.25 | p = 0.29 | p = 0.0077 | p = 1 | p < 0.0001 |
| Abnormalities in $K_{CO}$‡ | GLI-2017 | 552 (27.6%) | 4 (5.5%) | 35 (10.6%) | 97 (49.0%) | 7 (9.3%) | 409 (30.9%) |

ILD: Interstitial lung disease; $D_{LCO}$: Diffusing capacity of the lung for carbon monoxide; ECSC-93: European Community for Steel and Coal 1993 reference values; [2,4] GLI-2017: Global Lung function Initiative 2017 reference values; [6] $V_A$: Alveolar volume; $K_{CO}$: Transfer coefficient of the lung for carbon monoxide. Data are presented as n (%). McNemar's tests were used.

‡Abnormalities in indices if the z-score was strictly less than -1.645.

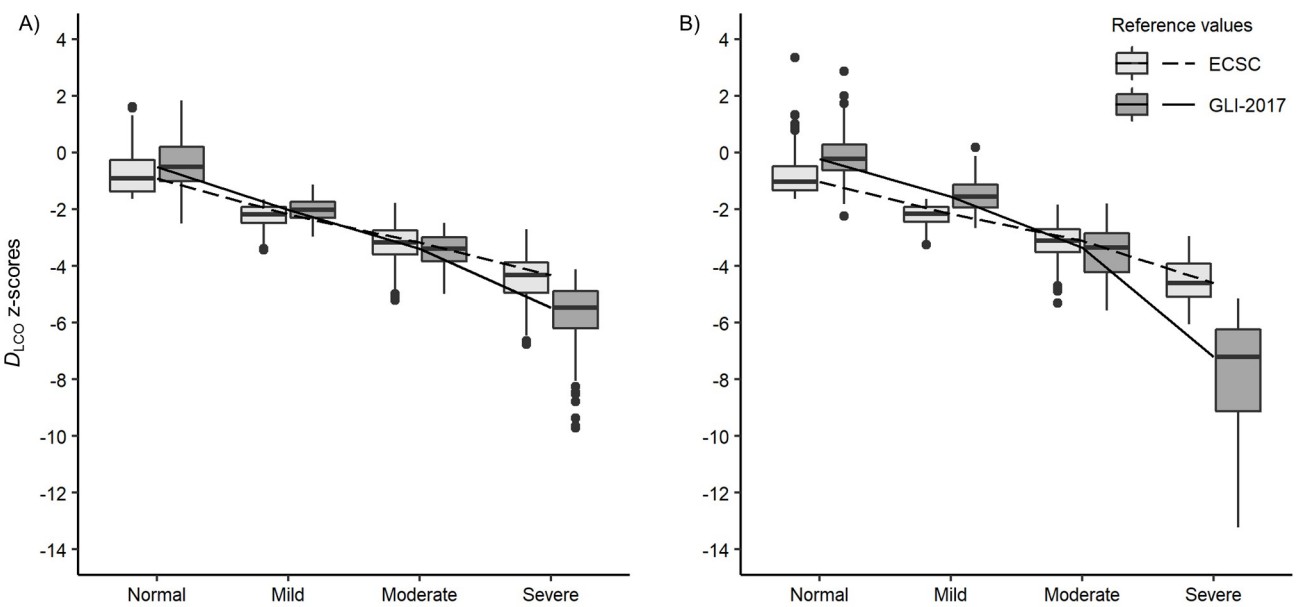

**Fig 2. Box-plots of $D_{LCO}$ ECSC-93 and GLI-2017 z-scores by degree of decrease in $D_{LCO}$ according to ATS/ERS-2005 in (A) men and (B) women with ILD.** Median values are represented by horizontal line within the boxes and interquartile ranges by box height. •: Outliers (values > 1.5 box lengths). $D_{LCO}$ was considered normal if the z-score was equal or greater than -1.645. The 2005 ATS/ERS recommendations for lung function test interpretation were used to define the degree of severity of $D_{LCO}$ impairment: "Mild" ($D_{LCO}$ > 60% predicted and below lower limits of normal), "Moderate" ($D_{LCO}$ > 40% and ≤ 60% predicted), "Severe" ($D_{LCO}$ ≤ 40% predicted) [18]. $D_{LCO}$: Diffusing capacity of the lung for carbon monoxide; ECSC: European Community for Steel and Coal; GLI-2017: Global Lung function Initiative 2017 reference values; [6] ATS/ERS-2005: 2005 recommendations for lung function test interpretation from the American Thoracic Society (ATS) and the European Respiratory Society (ERS) [18].

2017 z-scores of the 3 $D_{LCO}$ indices are closer to zero (expected value in a normal population) than ECSC-93 z-scores and the proportion of tests with an impaired $D_{LCO}$ is close to the expected 5%. This finding is important for clinical practice, obviating the need for additional diagnostic evaluations in females with falsely abnormal $D_{LCO}$ when using ECSC.

Our results are consistent with recent literature. In our study, GLI-2017 produce significantly higher %predicted than ECSC for $D_{LCO}$ and $V_A$ in both gender (all groups combined) with larger differences in women. These findings were already reported on mathematical modelling by Oostveen et al. [22]. Similarly to our results, Brazzale et al. [10] showed that GLI-2017 tend to increase the proportion of tests classified as normal, with a larger difference in younger women, compared to ECSC-93 and that the level of agreement between GLI and ECSC-93 was lower for females than for males. Unfortunately, there is no information about the pulmonary condition of the included subjects and the analyses were limited to the use of the LLN. Finally, Wapenaar et al. [9] have compared GLI-2017 to ECSC for $D_{LCO}$ in patient with idiopathic pulmonary fibrosis but their sample was composed of 82% of men and there was no analysis by sex.

In asthma and cystic fibrosis groups, GLI-2017 $D_{LCO}$ z-scores were significantly higher than ECSC z-scores in both genders. Conversely, in the chronic bronchitis and ILD groups, results are discordant between women and men. $D_{LCO}$ impairment severity seems to provide some clues. In the ILD group, Fig 2 shows an inversion of the trend for "moderate" or "severe" decrease in $D_{LCO}$: in "normal" $D_{LCO}$ or "mild" decrease in $D_{LCO}$, GLI-2017 produce significantly higher z-scores than ECSC, whereas, in "moderate" or "severe" decrease, GLI-2017

produce significantly lower z-scores than ECSC. This observation is even more important in women. Mathematical modelling showed an inversion of the GLI/ECSC z-scores ratio when $D_{LCO}$ is extremely decreased (see S1 Fig). These findings are consistent with recent literature [9].

One of the main strengths of our study is the size of our population. GLI-2017 were applied to 4180 tests (including 2898 subjects). To our knowledge, this is the first study to compare the GLI-2017 reference values to the ECSC using z-scores which should be favored in the interpretation of PFT results according to the ATS/ERS [1]. We also included control subjects, allowing us to applied the approach used for GLI-2012 [20,21]. To date, studies similar to ours has included patients without information about their pulmonary condition [10] or including only patients with a specific disease (e.g., idiopathic pulmonary fibrosis [9]). We have included PFTs from patients with symptoms compatible with one of four diseases (asthma, chronic bronchitis, cystic fibrosis or ILD). The chosen disease groups seemed to be the most relevant to us according to their definition and the diagnostic entities implied. It is worth noticing that patients with uncertain or associated diagnosis (e.g., a patient with asthma and chronic bronchitis) were excluded. Furthermore, all of the PFTs were performed on the same type of device, by a unique team, implying homogeneity in measurement method.

Our study also shows some weaknesses. We have compared the GLI-2017 reference values only to the ECSC in adults. The ECSC equations are currently one of the most used in European laboratories. Also, focusing on only one equation set and only on adult patients allowed us to detail different types of results (%predicted, z-scores and the 2005-ATS/ERS degrees of $D_{LCO}$ impairment [18]) and study those results in controls and multiple categories of patients with several degrees of decrease in $D_{LCO}$. This retrospective study uses highly suspected diagnoses, established by the referring physician (pulmonologist or internist) prescribing the PFT, and recorded in the laboratory database. However, in order to ensure data quality, diagnoses were checked from patient files in random samples of each groups. Some PFTs from the chronic bronchitis group are classified as normal (16.5% in women and 15% in men). There are 2 main reasons. First, we included tests performed at every stage of the diseases, including patients with mild symptoms of chronic bronchitis without obstruction. Secondly, we used z-score to define obstruction according to the 2005-ATS/ERS recommendations for lung function test interpretation [18] (z-score of $FEV_1/FVC$ ratio $< -1.645$ and TLC z-score $\geq -1.645$). Using the GOLD 2021 definition for COPD ($FEV_1/FVC < 0.7$) [16] could have led us to classify fewer PFTs as normal in the chronic bronchitis group in our sample due to the fact that a $FEV_1/FVC$ ratio $< 0.7$ can be associated with a z-score of the $FEV_1/FVC$ ratio $> -1.645$. On the other hand, some PFTs in the chronic bronchitis group were classified as restrictive without obstruction (3.1% in women and 6.8% in men with chronic bronchitis). This may happen in case of chronic bronchitis evaluation in a patient with a concomitant disease which has not been clinically detected by the prescribing physician. Also, we were unable to collect ethnicity, this data collection being strictly regulated by French law. However, the majority of the laboratory patients are Caucasians. We did not take into account active tobacco consumption and hemoglobinemia (which would only have corrected $D_{LCO}$ crude value). The aim of this study was not to assess the GLI-2017 diagnostic interest but to compare them to ECSC. Finally, our data come from a single laboratory, but it cover an area of more than 4 million people (approximately 9% of the French population) [23].

In conclusion, we showed that GLI-2017 reference values are more suitable to our sample than ECSC and that the differences between the 2 equation sets depend on the severity of $D_{LCO}$ impairment in the Interstitial Lung Disease group. Our study shows the impact of a possible future application of these new reference values on the diagnostic criteria for several lung diseases and on the definition of their severity.

## Supporting information

**S1 Fig. Mathematical modelling of theorical $D_{LCO}$ as a function of ECSC-93 and GLI-2017 z-scores and height in a 30 years old a) man and b) woman.** $D_{LCO}$: Diffusing capacity of the lung for carbon monoxide; ECSC: European Community for Steel and Coal 1993 reference values [2,4]; GLI: Global Lung function Initiative 2017 reference values [6].
(PDF)

**S1 Appendix. Quality control of the disease group classification.** Contains: S1 Table. Sample sizes, number of classification errors in the random sample and estimated classification error rate on the population of the study according to the ISO 2859–1 standard [24]. ILD: interstitial lung disease. Data are presented as n or %. §Classification error rate estimated for the overall population of the study in each disease group according to the ISO 2859–1 standard [24].
(PDF)

## Acknowledgments

The authors thank all of the staff of the Pulmonary Functional Tests Department for their help in this work.

## Author Contributions

**Conceptualization:** Pierre-Marie Wardyn, Virginie de Broucker, Jean-Louis Edmé, Sébastien Hulo.

**Data curation:** Virginie de Broucker, Richard Van Bulck, Jean-Louis Edmé.

**Formal analysis:** Pierre-Marie Wardyn, Jean-Louis Edmé.

**Investigation:** Richard Van Bulck, Jean-Louis Edmé.

**Methodology:** Pierre-Marie Wardyn, Virginie de Broucker, Jean-Louis Edmé, Sébastien Hulo.

**Project administration:** Jean-Louis Edmé, Sébastien Hulo.

**Supervision:** Jean-Louis Edmé, Sébastien Hulo.

**Validation:** Pierre-Marie Wardyn, Virginie de Broucker, Cécile Chenivesse, Annie Sobaszek, Richard Van Bulck, Thierry Perez, Jean-Louis Edmé, Sébastien Hulo.

**Writing – original draft:** Pierre-Marie Wardyn, Virginie de Broucker, Jean-Louis Edmé, Sébastien Hulo.

**Writing – review & editing:** Pierre-Marie Wardyn, Virginie de Broucker, Cécile Chenivesse, Annie Sobaszek, Richard Van Bulck, Thierry Perez, Jean-Louis Edmé, Sébastien Hulo.

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
