## [Editor Report · Decision Letter 0]

28 Aug 2020

PONE-D-20-24269

Application of the new Global Lung Function Initiative reference values for the diffusing capacity of the lung for carbon monoxide to a large population

PLOS ONE

Dear Dr. HULO,

Thank you for submitting your manuscript to PLOS ONE. After careful consideration, we feel that it has merit but does not fully meet PLOS ONE’s publication criteria as it currently stands. Therefore, we invite you to submit a revised version of the manuscript that addresses the points raised during the review process.

I would recommend the change of the title of the manuscript to: „Assessing the applicability of the new Global Lung Function Initiative reference values for the diffusing capacity of the lung for carbon monoxide in a large population set“

Please describe in detail how the used tests were evaluated for inclusion/exclusion into the study.

Add actual p-values to Table 1. You mentioned that “GLI-2012 produce significantly lower median z-scores for FEV1, FVC and FEV1/FVC 180 than ECSC, all groups combined, in both genders (table 2).” but no p-values were cited. Please include actual p-values in the Results. Also for Table 3 include actual p-values and refer them in the text where you state significant differences. Please do the same for Table 4.

We look forward to receiving your revised manuscript.

Kind regards,

Davor Plavec, MD, MSc, PhD, Prof.

Academic Editor

PLOS ONE

Additional Editor Comments:

I would recommend the change of the title of the manuscript to: „Assessing the applicability of the new Global Lung Function Initiative reference values for the diffusing capacity of the lung for carbon monoxide in a large population set“

Please describe in detail how the used tests were evaluated for inclusion/exclusion into the study.

Add actual p-values to Table 1. You mentioned that “GLI-2012 produce significantly lower median z-scores for FEV1, FVC and FEV1/FVC 180 than ECSC, all groups combined, in both genders (table 2).” but no p-values were cited. Please include actual p-values in the Results. Also for Table 3 include actual p-values and refer them in the text where you state significant differences. Please do the same for Table 4.
---

## [Author Response · Author response to Decision Letter 0]

10 Oct 2020

EDITOR COMMENTS:

1) I would recommend the change of the title of the manuscript to: „Assessing the applicability of the new Global Lung Function Initiative reference values for the diffusing capacity of the lung for carbon monoxide in a large population set“

The title was changed from “Application of the new Global Lung Function Initiative reference values for the diffusing capacity of the lung for carbon monoxide to a large population” to „Assessing the applicability of the new Global Lung Function Initiative reference values for the diffusing capacity of the lung for carbon monoxide in a large population set“

2) Please describe in detail how the used tests were evaluated for inclusion/exclusion into the study.

In order to describe in detail how the used tests were evaluated for inclusion/exclusion into the study, the following paragraph was added in “materiel and methods”, in the “study population” part:

“For each test, PFTs indices values and patient’s characteristics (sex, age, height, weight) are recorded in the laboratory database and a test report including patient’s medical history is created using Microsoft Access® which is also recorded in the laboratory database. For the tests’ inclusion, data from the PFT performed between November 15th 2012 and May 30th 2016, and including the indices mentioned above, were extracted to a Microsoft Excel® spreadsheet by the laboratory engineer. Then tests meeting exclusion criteria were excluded using the statistical software SAS® (version 9.4; Statistical Analysis System).”

3) Add actual p-values to Table 1. You mentioned that “GLI-2012 produce significantly lower median z-scores for FEV1, FVC and FEV1/FVC 180 than ECSC, all groups combined, in both genders (table 2).” but no p-values were cited. Please include actual p-values in the Results. Also for Table 3 include actual p-values and refer them in the text where you state significant differences. Please do the same for Table 4.

Actual p-values were added to Table 1, 3 and 4. Actual p-values were added in the Results. We set the limit to p < 0.0001.

---

## [Decision Letter · Decision Letter 1]

10 Nov 2020

PONE-D-20-24269R1

Assessing the applicability of the new Global Lung Function Initiative reference values for the diffusing capacity of the lung for carbon monoxide in a large population set

PLOS ONE

Dear Dr. HULO,

Thank you for submitting your manuscript to PLOS ONE. After careful consideration, we feel that it has merit but does not fully meet PLOS ONE’s publication criteria as it currently stands. Therefore, we invite you to submit a revised version of the manuscript that addresses the points raised during the review process.

Please do the suggested minor revision by the reviewer or write a detailed rebuttal prior to publication. 

We look forward to receiving your revised manuscript.

Kind regards,

Davor Plavec, MD, MSc, PhD, Prof.

Academic Editor

PLOS ONE

Additional Editor Comments (if provided):

Please do the suggested minor revision by the reviewer or write a detailed rebuttal prior to publication.

Reviewers' comments:

Reviewer's Responses to Questions

**Comments to the Author**

1. If the authors have adequately addressed your comments raised in a previous round of review and you feel that this manuscript is now acceptable for publication, you may indicate that here to bypass the “Comments to the Author” section, enter your conflict of interest statement in the “Confidential to Editor” section, and submit your "Accept" recommendation.

Reviewer #1: (No Response)

2. Is the manuscript technically sound, and do the data support the conclusions?

Reviewer #1: Partly

3. Has the statistical analysis been performed appropriately and rigorously? 

Reviewer #1: I Don't Know

4. Have the authors made all data underlying the findings in their manuscript fully available?

Reviewer #1: Yes

5. Is the manuscript presented in an intelligible fashion and written in standard English?

Reviewer #1: Yes

6. Review Comments to the Author

Reviewer #1: For my point of view there is inconsistency in the definition of the primary disease. Was it "patients with symptoms compatible with one of the diseases" or clinical diagnosis (confirmed or highly suspected diagnoses, established by the referring physician) or "diagnosis were checked from patient files in random samples of each groups"?

That should be better characterised and if the disease was not properly validated the groups should be named accordingly.

The 15% of normal lung function in the COPD group and 7% restriction without obstruction should be explained in the methods and addressed in the discussion.

7. PLOS authors have the option to publish the peer review history of their article (what does this mean?). If published, this will include your full peer review and any attached files.

Reviewer #1: No

---

## [Author Response · Author response to Decision Letter 1]

25 Nov 2020

EDITOR COMMENTS:

Please do the suggested minor revision by the reviewer or write a detailed rebuttal prior to publication.

Please find below our responses to the suggested minor revision by the reviewer.

REVIEWERS' COMMENTS:

2. Is the manuscript technically sound, and do the data support the conclusions?

Reviewer #1: Partly

In our study, we aimed not to verify the interest of measuring DLCO for specific pulmonary diseases but to assess the applicability of the GLI-2017 reference values for DLCO of adult subjects by comparing the GLI-2017 reference values to the ECSC-93 equations currently used in a large population set. The data in this study support our conclusions: the GLI-2017 reference values are more suitable to our population than the ECSC-93 equations and differences between the 2 equations sets depend on the severity of DLCO impairment in the Interstitial Lung Disease group.

For my point of view there is inconsistency in the definition of the primary disease. Was it "patients with symptoms compatible with one of the diseases" or clinical diagnosis (confirmed or highly suspected diagnoses, established by the referring physician) or "diagnosis were checked from patient files in random samples of each groups"? That should be better characterised and if the disease was not properly validated the groups should be named accordingly.

The diagnoses were confirmed or highly suspected, established by the referring physician (pulmonologist or internist) prescribing the PFT, and to ensure data quality, we checked the diagnoses from patient files in random samples of each groups.

This point was clarified in materials and methods (“In order to ensure data quality, diagnoses were checked from patient files in random samples of each groups”) and discussion (“This retrospective study uses confirmed or highly suspected diagnoses, established by the referring physician (pulmonologist or internist) prescribing the PFT, and recorded in the laboratory database. However, in order to ensure data quality, diagnoses were checked from patient files in random samples of each groups.”).

The 15% of normal lung function in the COPD group and 7% restriction without obstruction should be explained in the methods and addressed in the discussion.

The fact that there are 15% of normal lung function in the COPD group and 7% restriction without obstruction have been explained in materials and methods (“Selected PFTs could have been performed at any stage of the concerned disease, from initial evaluation to advanced stage”) and addressed in the discussion (“We included tests performed at every stage of the diseases, which can be very early, explaining why some PFTs from patients with COPD are classified as normal (16.5% in women and 15% in men). On the other hand, some are classified as restrictive without obstruction (3.1% in women and 6.8% in men with COPD). This can append in case of an early stage COPD evaluation in a patient with a concomitant disease which has not been clinically detected by the prescribing physician.”)

It is also worth noticing that we aimed not to verify the interest of measuring DLCO for specific pulmonary diseases but to assess the applicability of the GLI-2017 reference values for DLCO of adult subjects.

Hoping this revised version of our work will meet with your approval,

Sincerely yours,

Pierre-Marie WARDYN, MD, Virginie de BROUCKER, MD, Cécile CHENIVESSE, MD, PhD, Annie SOBASZEK, MD, PhD, Richard VAN BULCK, Thierry PEREZ, MD, Jean-Louis EDME, PhD, Sébastien HULO, MD, PhD.

---

## [Decision Letter · Decision Letter 2]

21 Dec 2020

PONE-D-20-24269R2

Assessing the applicability of the new Global Lung Function Initiative reference values for the diffusing capacity of the lung for carbon monoxide in a large population set

PLOS ONE

Dear Dr. HULO,

Thank you for submitting your manuscript to PLOS ONE. After careful consideration, we feel that it has merit but does not fully meet PLOS ONE’s publication criteria as it currently stands. Therefore, we invite you to submit a revised version of the manuscript that addresses the points raised during the review process.

Please resolve the issue raised by the reviewer.

We look forward to receiving your revised manuscript.

Kind regards,

Davor Plavec, MD, MSc, PhD, Prof.

Academic Editor

PLOS ONE

Additional Editor Comments (if provided):

Dear Authors,

please resolve the issue raised by the reviewer.

Reviewers' comments:

Reviewer's Responses to Questions

**Comments to the Author**

1. If the authors have adequately addressed your comments raised in a previous round of review and you feel that this manuscript is now acceptable for publication, you may indicate that here to bypass the “Comments to the Author” section, enter your conflict of interest statement in the “Confidential to Editor” section, and submit your "Accept" recommendation.

Reviewer #1: (No Response)

2. Is the manuscript technically sound, and do the data support the conclusions?

Reviewer #1: Partly

3. Has the statistical analysis been performed appropriately and rigorously? 

Reviewer #1: I Don't Know

4. Have the authors made all data underlying the findings in their manuscript fully available?

Reviewer #1: Yes

5. Is the manuscript presented in an intelligible fashion and written in standard English?

Reviewer #1: Yes

6. Review Comments to the Author

Reviewer #1: This is the explanation of my decision and what I believe should be corrected:

Your explanation: “The fact that there are 15% of normal lung function in the COPD group and 7%

restriction without obstruction have been explained in materials and methods

(“Selected PFTs could have been performed at any stage of the concerned disease,

from initial evaluation to advanced stage”) and addressed in the discussion (“We

included tests performed at every stage of the diseases, which can be very early,

explaining why some PFTs from patients with COPD are classified as normal (16.5% in

women and 15% in men). On the other hand, some are classified as restrictive without

obstruction (3.1% in women and 6.8% in men with COPD).”

GOLD 2021: The FEV1/FVC<0,70 is required for the diagnosis of CPOD, normal lung function or restriction without obstruction are ruling out COPD as diagnosis. Your explanation is not in accordance with actual guidelines. It does not matter whether it is important to the whole article, but this group can not be declared as COPD patients.

Study population

We selected data of a control group and patients with symptoms compatible with one of four diseases* (asthma, chronic obstructive pulmonary disease (COPD), cystic fibrosis, and interstitial lung diseases (ILD))

*not with the diagnosed disease!

Groups creation

Selected PFTs came from 4 groups of patients with symptoms compatible* with one of the following diseases: asthma, COPD, cystic fibrosis, and ILD.

*not with the diagnosed disease!

Discussion

We have included PFTs from patients with one defined disease* (asthma, COPD, cystic fibrosis or ILD).

*now we have a defined disease?

This retrospective study uses confirmed or highly suspected diagnoses*, established by the referring physician (pulmonologist or internist) prescribing the PFT.

*Here we have confirmed or highly suspected disease!

We included tests performed at every stage of the diseases, which can be very early, explaining why some PFTs from patients with COPD are classified as normal* (16.5% in women and 15% in men).

*According to actual guidelines, there is no early COPD with normal lung function or please add a reference for that

On the other hand, some are classified as restrictive without obstruction (3.1% in women and 6.8% in men with COPD). This can append* in case of an early-stage COPD** evaluation in a patient with a concomitant disease which has not been clinically detected by the prescribing physician.

*append, maybe happened?

**please correct or reference

Dear colleagues,

I value your work and believe that it is worth of publishing but not with the wrong definition of the COPD. I would suggest you correct the problems I have note with the asterisks. I see two solutions: to rename the COPD group or to exclude the patients without obstruction from the statistical analyze of COPD group.

Best regards

7. PLOS authors have the option to publish the peer review history of their article (what does this mean?). If published, this will include your full peer review and any attached files.

Reviewer #1: No

---

## [Author Response · Author response to Decision Letter 2]

30 Dec 2020

Dear Editor,

Please find below the responses to each point raised during the reviewing process of our manuscript entitled “Assessing the applicability of the new Global Lung Function Initiative reference values for the diffusing capacity of the lung for carbon monoxide in a large population set” (authors: P.-M. WARDYN, V. de BROUCKER, C. CHENIVESSE, A. SOBASZEK, R. VAN BULCK, T. PEREZ, J.-L. EDME, S. HULO):

EDITOR COMMENTS:

Please resolve the issue raised by the reviewer.

Please find below the solution we proposed in order to resolve the issue raised by the reviewer.

REVIEWERS' COMMENTS:

We would like to thank the reviewer for his interest in our work.

This is the explanation of my decision and what I believe should be corrected:

Your explanation: “The fact that there are 15% of normal lung function in the COPD group and 7% restriction without obstruction have been explained in materials and methods (“Selected PFTs could have been performed at any stage of the concerned disease, from initial evaluation to advanced stage”) and addressed in the discussion (“We included tests performed at every stage of the diseases, which can be very early, explaining why some PFTs from patients with COPD are classified as normal (16.5% in women and 15% in men). On the other hand, some are classified as restrictive without obstruction (3.1% in women and 6.8% in men with COPD).”

GOLD 2021: The FEV1/FVC<0,70 is required for the diagnosis of CPOD, normal lung function or restriction without obstruction are ruling out COPD as diagnosis. Your explanation is not in accordance with actual guidelines. It does not matter whether it is important to the whole article, but this group can not be declared as COPD patients. 

In order to clarify our definition of our “COPD group” and to be in accordance with the GOLD 2021, we opted for your second proposition mentioned below: we replaced the name of the "COPD" group with the term "chronic bronchitis". We included the following sentence in the methods in the “group creation” part: “The chronic bronchitis group included patients with symptoms compatible with chronic bronchitis (with a FEV1/FVC ratio ≥ 0.7) or with chronic obstructive pulmonary disease (COPD) (with a FEV1/FVC ratio < 0.7 according to the GOLD 2021 report[1])”.

Study population

We selected data of a control group and patients with symptoms compatible with one of four diseases* (asthma, chronic obstructive pulmonary disease (COPD), cystic fibrosis, and interstitial lung diseases (ILD)) 

*not with the diagnosed disease!

We harmonized the text with the formulation “patients with symptoms compatible with […]” and replaced COPD by chronic bronchitis, and so, we modified the sentence as follow: “We selected data of a control group and patients with symptoms compatible with with one of four diseases* (asthma, chronic bronchitis, cystic fibrosis, and interstitial lung diseases (ILD))”.

Groups creation

Selected PFTs came from 4 groups of patients with symptoms compatible* with one of the following diseases: asthma, COPD, cystic fibrosis, and ILD.

*not with the diagnosed disease!

We harmonized the text with the formulation “patients with symptoms compatible with […]” and replaced COPD by chronic bronchitis, and so, we modified the sentence as follow: “Selected PFTs came from 4 groups of patients with symptoms compatible with one of the following diseases: asthma, chronic bronchitis, cystic fibrosis, and ILD.”

Discussion

We have included PFTs from patients with one defined disease* (asthma, COPD, cystic fibrosis or ILD).

*now we have a defined disease?

We modified the formulation by “patients with symptoms compatible with one of four diseases (asthma, chronic bronchitis, cystic fibrosis or ILD)” in order to harmonize the text of the discussion section with the methods section mentioned above.

This retrospective study uses confirmed or highly suspected diagnoses*, established by the referring physician (pulmonologist or internist) prescribing the PFT.

*Here we have confirmed or highly suspected disease!

We modified the formulation by “This retrospective study uses highly suspected diagnoses […]”.

We included tests performed at every stage of the diseases, which can be very early, explaining why some PFTs from patients with COPD are classified as normal* (16.5% in women and 15% in men).

*There is no early COPD with normal lung function or add a reference

The sentence was modified by “Some PFTs from the chronic bronchitis group are classified as normal (16.5% in women and 15% in men). There are 2 main reasons. First, we included tests performed at every stage of the diseases, including patients with mild symptoms of chronic bronchitis without obstruction. Secondly, we used z-score to define obstruction according to the 2005-ATS/ERS recommendations for lung function test interpretation[2] (z-score of FEV1/FVC ratio < -1.645 and TLC z-score ≥ -1.645). Using the GOLD 2021 definition for COPD (FEV1/FVC < 0.7)[1] could have led us to classify fewer PFTs as normal in the chronic bronchitis group in our sample due to the fact that a FEV1/FVC ratio < 0.7 can be associated with a z-score of the FEV1/FVC ratio > -1.645”.

On the other hand, some are classified as restrictive without obstruction (3.1% in women and 6.8% in men with COPD). This can append* in case of an early-stage COPD** evaluation in a patient with a concomitant disease which has not been clinically detected by the prescribing physician.

*append, maybe happened?

This typing error was corrected.

**please correct or reference

The sentence was changed to “On the other hand, some PFTs in the chronic bronchitis group were classified as restrictive without obstruction (3.1% in women and 6.8% in men with chronic bronchitis). This may happen in case of chronic bronchitis evaluation in a patient with a concomitant disease which has not been clinically detected by the prescribing physician”.

I value your work and believe that it is worth of publishing but not with the wrong definition of the COPD. I would suggest you correct the problems I have note with the asterisks. I see two solutions: to rename the COPD group or to exclude the patients without obstruction from the statistical analyze of COPD group.

As mentioned above, we renamed the COPD group as the chronic bronchitis group.

Hoping this revised version of our work will meet with your approval,

Sincerely yours,

Pierre-Marie WARDYN, MD, Virginie de BROUCKER, MD, Cécile CHENIVESSE, MD, PhD, Annie SOBASZEK, MD, PhD, Richard VAN BULCK, Thierry PEREZ, MD, Jean-Louis EDME, PhD, Sébastien HULO, MD, PhD.

REFERENCES

1. Global initiative for chronic obstructive lung disease (GOLD). Global strategy for the diagnosis, management, and prevention of chronic obstructive pulmonary disease (2021 report). 2020. Available: https://goldcopd.org/2021-gold-reports/

2. Pellegrino R, Viegi G, Brusasco V, Crapo RO, Burgos F, Casaburi R, et al. Interpretative strategies for lung function tests. European Respiratory Journal. 2005;26: 948–968. doi:10.1183/09031936.05.00035205

---

## [Decision Letter · Decision Letter 3]

4 Jan 2021

Assessing the applicability of the new Global Lung Function Initiative reference values for the diffusing capacity of the lung for carbon monoxide in a large population set

PONE-D-20-24269R3

Dear Dr. HULO,

We’re pleased to inform you that your manuscript has been judged scientifically suitable for publication and will be formally accepted for publication once it meets all outstanding technical requirements.

Kind regards,

Davor Plavec, MD, MSc, PhD, Prof.

Academic Editor

PLOS ONE

Additional Editor Comments (optional):

After the last revision the reviewer suggested accepting your manuscript.

Reviewers' comments:

Reviewer's Responses to Questions

**Comments to the Author**

1. If the authors have adequately addressed your comments raised in a previous round of review and you feel that this manuscript is now acceptable for publication, you may indicate that here to bypass the “Comments to the Author” section, enter your conflict of interest statement in the “Confidential to Editor” section, and submit your "Accept" recommendation.

Reviewer #1: All comments have been addressed

2. Is the manuscript technically sound, and do the data support the conclusions?

Reviewer #1: Yes

3. Has the statistical analysis been performed appropriately and rigorously? 

Reviewer #1: I Don't Know

4. Have the authors made all data underlying the findings in their manuscript fully available?

Reviewer #1: Yes

5. Is the manuscript presented in an intelligible fashion and written in standard English?

Reviewer #1: Yes

6. Review Comments to the Author

Reviewer #1: (No Response)

7. PLOS authors have the option to publish the peer review history of their article (what does this mean?). If published, this will include your full peer review and any attached files.

Reviewer #1: No

---

## [Editor Report · Acceptance letter]

5 Jan 2021

PONE-D-20-24269R3 

Assessing the applicability of the new Global Lung Function Initiative reference values for the diffusing capacity of the lung for carbon monoxide in a large population set 

Dear Dr. HULO:

I'm pleased to inform you that your manuscript has been deemed suitable for publication in PLOS ONE. Congratulations! Your manuscript is now with our production department. 

Kind regards, 

on behalf of

Dr. Davor Plavec 

Academic Editor

PLOS ONE